# Clinical Progress in Proton Radiotherapy: Biological Unknowns

**DOI:** 10.3390/cancers13040604

**Published:** 2021-02-03

**Authors:** Laura Vanderwaeren, Rüveyda Dok, Kevin Verstrepen, Sandra Nuyts

**Affiliations:** 1Laboratory of Experimental Radiotherapy, Department of Oncology, KU Leuven, 3000 Leuven, Belgium; laura.vanderwaeren@kuleuven.be (L.V.); ruveyda.dok@kuleuven.be (R.D.); 2Laboratory of Genetics and Genomics, Centre for Microbial and Plant Genetics, KU Leuven, 3000 Leuven, Belgium; kevin.verstrepen@kuleuven.be; 3Laboratory for Systems Biology, VIB-KU Leuven Center for Microbiology, 3000 Leuven, Belgium; 4Department of Radiation Oncology, Leuven Cancer Institute, University Hospitals Leuven, 3000 Leuven, Belgium

**Keywords:** proton radiation, radiobiology, radiotherapy

## Abstract

**Simple Summary:**

Proton radiation therapy is a more recent type of radiotherapy that uses proton beams instead of classical photon or X-rays beams. The clinical benefit of proton therapy is that it allows to treat tumors more precisely. As a result, proton radiotherapy induces less toxicity to healthy tissue near the tumor site. Despite the experience in the clinical use of protons, the response of cells to proton radiation, the radiobiology, is less understood. In this review, we describe the current knowledge about proton radiobiology.

**Abstract:**

Clinical use of proton radiation has massively increased over the past years. The main reason for this is the beneficial depth-dose distribution of protons that allows to reduce toxicity to normal tissues surrounding the tumor. Despite the experience in the clinical use of protons, the radiobiology after proton irradiation compared to photon irradiation remains to be completely elucidated. Proton radiation may lead to differential damages and activation of biological processes. Here, we will review the current knowledge of proton radiobiology in terms of induction of reactive oxygen species, hypoxia, DNA damage response, as well as cell death after proton irradiation and radioresistance.

## 1. Introduction

Radiation is a crucial component in the treatment of cancer. The first cancer patient treated with radiation therapy dates back to the end of the 19th century [1,2]. Today, for up to 50% of cancer patients radiotherapy is part of their treatment [3]. The oldest, classic irradiation technique, namely, X-ray or photon therapy, underwent many technical improvements in accuracy and planning over the years. As a result, the ratio between the optimal dose to the tumor and the lowest possible dose to the surrounding tissue improved. However, toxicity problems for the healthy tissue surrounding the tumor remain a major problem in conventional X-ray or photon radiotherapy because of the relatively high entrance and exit dose of photons, as is visualized in Figure 1 [4]. This limits the ability to increase the dose administered to the tumor while preserving normal tissue. The latter becomes especially important when high-risk organs are located in the vicinity of the tumor site.

A newer irradiation technique, namely, proton radiotherapy, led to an expansion in the treatment possibilities of tumors. Proton therapy enables a better depth dose distribution compared to the technique of classical X-rays. Protons are charged particles for which the range of penetration into tissue is well defined. While penetrating tissues, protons are slowed down and deposit most of their energy at the end of their range. This results in a very characteristic depth-dose distribution called the Bragg peak. Modifying the energy of the accelerated protons, the range of penetration can be adjusted. That allows to combine multiple Bragg peaks into a so-called spread-out-Bragg peak (SOBP) that encompasses the whole tumor, while limiting the dose received to normal tissue, as depicted in Figure 1.

The use of protons for clinical applications in cancer treatment was first suggested by physicist Robert R. Wilson in 1946 [5]. However, it took until 1954 before the first patients were treated with proton radiotherapy [6]. Because of technical limitations in the accelerators, protons could not reach a high enough energy to penetrate the body for tumor treatment. Therefore, the early applications of proton therapy were limited to superficially located tumors [7,8,9,10]. The combination of research from the initial proton radiotherapy facilities as well as technical progression and improvements in terms of compactness and cost reduction paved the way for proton therapy [11]. More and more cyclotrons are built to establish proton therapy centers worldwide and make the therapy accessible to a variety of clinical indications. Currently, 98 proton therapy facilities are clinically operational according to the Particle Therapy Co-Operative Group. An additional 28 therapy centers are in the planning stage and 31 more are under construction [12]. Almost all of these proton therapy facilities became operational in the last 10 years, highlighting the clinical need and progress of protons in cancer treatment. Although proton therapy is now widely used to treat numerous cancer patients, there are still major hurdles to overcome in terms of physical delivery as well as unknowns about the radiobiology of protons.

Even though the theoretical advantage of protons is apparent, these same physical properties can also pose a risk due to uncertainties about the in vivo range of protons caused by an interplay of not only setup errors, CT artefacts, uncertainties in stopping power measurements, and conversion but also patient motion and anatomical changes during treatment [13,14,15]. An underestimation of safety margins in photon therapy might cause underdosage of the tumor, while in proton therapy, this can lead to part of the tumor not receiving any dose due to a shift of the sharp distal dose fall-off. However, more importantly, a shift of the Bragg peak can cause extensive normal tissue toxicity when positioned incorrectly, which is less so for photon therapy. As a result, in clinical practice, considerable safety margins and conservative planning strategies are generally applied [15]. These problems in physical delivery of protons and how to tackle these are not discussed in further detail, and the reader is referred to recent reviews on this topic [15,16,17,18,19].

In this review paper, we will focus on the biological unknowns of proton therapy as the radiobiological research is falling behind on the clinical research. Now, with the increasing amount of proton therapy facilities with integrated radiobiological research centers, more and more radiobiological research is focused towards protons in comparison with conventional photons. Therefore, in this review, an overview will be given about recent advances in understanding the radiobiology of protons, and we will focus on why this is important to further improve the clinical treatments. First, the ongoing discussion about the relative biological effectiveness (RBE) of protons will be discussed. Next, we will give an overview of recent research about the biological processes involved in the response to proton irradiation. The importance of each of these fields related to clinical therapies will be highlighted and research gaps will be discussed.

## 2. The Ongoing Debate about the RBE of Protons

Relative biological effectiveness (RBE) is a concept used to compare the biological effectiveness of a certain type of radiation to a reference radiation type, usually photons. The proton RBE is then defined as the ratio of the dose of photons to the dose of protons that is needed to cause the same biological effect [20]. In clinical settings, a generic constant RBE of 1.1 is applied in proton therapy planning. However, the RBE depends on physical properties such as the proton beam energy, the dose and dose rate, the fractionation scheme, and the position of irradiation along the Bragg peak profile but is also influenced by biological properties such as the tissue type or cell line, the cell cycle stage, the oxygenation level, and the studied endpoint [21,22,23,24].

The linear energy transfer (LET), defined as the amount of energy deposited per unit distance along the particles track, is one important factor that influences the RBE [20]. As particles, such as protons or carbon ions, are slowed down, they lose more of their energy and a maximum is reached at the distal edge of the Bragg peak. As LET is a measure of ionization density, increasing LET leads to denser ionization events along the particle track. This eventually results in more extensive and more clustered biological damage [25]. Indeed, multiple studies have shown that the RBE of protons increases along the Bragg peak profile [26,27,28,29]. A comprehensive review of 2014 reported an average RBE for cell survival of 1.1 at the entrance, 1.15 in the center, to 1.35 at the distal edge, and up to 1.7 in the distal fall-off of the SOBP [21]. However, higher RBE values have also been reported [30,31,32,33,34]. For high LET radiation like carbon ions, which are particle beams like protons, RBEs up to 3 have been reported [35].

Even though there is increasing evidence that the proton RBE varies along the SOBP, the debate about whether or not to change the current clinical practice, which uses the fixed RBE of 1.1, is still ongoing [36,37,38,39,40,41]. The use of this constant RBE for protons is often justified with the uncertainties in the available RBE data. There remains considerable variety in the reported in vitro RBEs, which is caused by changing experimental conditions as well as differences in cell biology. The dependency of the RBE on the physical properties is more accepted as a systematic effect. The dependency on the biological properties is understudied. This severely limits to reach the true potential of proton therapy. Therefore, the intricate interplay between the biological processes important after proton therapy needs to be more understood.

## 3. Reactive Oxygen Species and Hypoxia

Ionizing radiation has both direct and indirect effects on the DNA resulting in DNA damage. Direct effects are caused by the radiation directly interacting with the atoms of the DNA resulting in DNA damage. Indirect effects are caused by the interaction of the radiation with water molecules, which results in the production of reactive oxygen species (ROS). These ROS can indirectly cause damage to the DNA by reacting with molecular oxygen to form stable DNA peroxides. Since the DNA molecules take up such a small part inside a cell, the probability of the radiation hitting the DNA is very low. It is estimated that 30–40% of DNA lesions are caused by direct interaction and the remaining 60–70% by indirect interaction [42]. Therefore, DNA damage caused by the indirect action of radiation is crucial for its effect. Consequently, cells are more sensitive to radiation when oxygen is present and, as a result, tumor hypoxia can lead to radioresistance [43,44]. However, this effect is smaller when cells are irradiated with higher LET radiation, because of the denser ionization events that cause clustered DNA damage. This can also be observed by comparing the oxygen enhancement ratio (OER), defined as the dose needed in hypoxic conditions divided by the dose needed in normoxic conditions to cause the same biological effect [45]. The OER decreases with increasing LET and ranges from 2.5 to 3 for photons and protons and 1.6 to 2 for carbon ions [46]. The OER for photons and protons is similar, thus carbon ions might be more suited to overcome the radioresistance of hypoxic tumors. However, protons also have some benefits compared to conventional photons in the treatment of hypoxic tumors related to the expression of HIF-1α. HIF-1α is a gene important in the radioresistance of hypoxic tumors that, when activated, increases the expression of genes involved in growth, energy metabolism, endothelial cell function, and neovascularization [47,48,49]. Induction of HIF-1α thus promotes tumor growth and can eventually lead to metastasis [50]. It has been shown that photon irradiation induces the expression of HIF-1α [47,51,52,53]. Although only studied by one research group, irradiation with 0.5, 1, and 2 Gy 1 GeV protons, on the other hand, decreases HIF-1α expression in different cell types compared to nonirradiated controls. In the same cell types, a dose-dependent increase in expression of HIF-1α was observed after similar doses of photons [52]. As this is the first report about a decreased HIF-1α expression after proton irradiation, more studies are needed to be able to validate the results.

Moreover, several reports have shown that proton therapy induces more ROS compared to photon therapy, which could be a big contributor to the increased RBE of protons [54,55,56,57,58]. Giedzinksi et al. [54] observed increased ROS levels in neural precursor cells after exposure to 250 MeV Bragg peak protons compared to X-rays [54]. Additionally, they also reported a more rapid and dose-dependent increase in ROS after proton radiation [54]. Similarly, in cancer stem-like cells, a dose-dependent increase in ROS was seen after irradiation with photons and protons. For proton irradiation, the cells were positioned at the center of a SOBP. The ROS levels in cells irradiated with 4 Gy protons were significantly higher than those after irradiation with the same dose of photons. Furthermore, it was suggested that the greater generation of ROS after protons induces more apoptosis in cancer stem cells compared to photons [58]. Mitteer et al. also found that irradiation of glioma stem cells with 5 or 10 Gy protons led to higher ROS induction than similar photon doses [57].

## 4. The DNA Damage Response after Proton Radiation

### 4.1. Induction of DNA Damage

When comparing the effect of photon and proton radiation, one parameter that differs between the two types of irradiation and that can influence the type of DNA damage that is induced is the LET. As already mentioned, increasing LET results in denser ionization events, which causes more extensive and more clustered DNA damage [25]. These clustered DNA damages can comprise several and different DNA lesions caused by a single track of ionizing radiation within a short DNA region of a few base pairs [59]. They can include single strand breaks (SSB), double strand breaks (DSB), abasic sites and base damages [25]. There are thus two basic groups of clustered DNA damage: DSBs and non-DSBs. A clustered DNA damage site can be composed of, e.g., a DSB in close proximity to an oxidative damage site, a DSB a few base pairs away from a SSB, and two SSBs or two base damages in close proximity. The presence of clustered DNA lesions after proton therapy is expected from Monte Carlo simulations, however, the detection of lesions in situ proves to be very difficult [25,60,61,62]. For an in-depth overview of detection methods for clustered DNA damage, the reader is referred to recent reviews on this topic [59,63]. It is worth highlighting that, recently, the group of Xu et al. [64] succeeded in visualizing clustered DNA damage by atomic force microscopy thereby providing the first experimental evidence for clustered DNA damage [64]. Because an easy, efficient direct method to detect clustered DNA damage is still lacking, evidence of the presence of these clustered lesions is mainly indirect, e.g., through colocalization immunofluorescent stainings or comet assays. However, the immunofluorescent studies are complicated by the presence of multiple DNA repair enzymes in close proximity at the site of these clustered regions. It then becomes necessary to precisely define when the detected proteins are indeed present at one site of clustered DNA damage, which requires well-equipped microscopes. As a result, even though there is evidence for proton-induced clustered DNA damage as a result of the increasing LET, papers describing the induction of DNA damage after proton radiation are in most cases not really measuring clustered DNA damage but rather DNA damage in general. Most studies thereby focus on the immunofluorescent detection of γH2AX foci. However, immunofluorescent staining of γH2AX will not determine if the detected DSB is part of a clustered DNA damage site. γH2AX is the phosphorylated variant of histone H2AX. Histones are proteins that associate with the DNA in order to pack and organize the DNA so that it fits inside the nucleus. As part of the initial DNA damage response to DSBs, a phosphoryl group is transferred to histone H2AX in a process called phosphorylation. This phosphorylation event happens quickly after a DSB is induced. It marks the site of the DSB and functions as a signal for repair [65,66]. Consequently, the detection of γH2AX foci is used to study the amount of DNA DSBs, and by following the removal of γH2AX foci over time, the repair kinetics can also be studied.

Comparing the distribution of γH2AX foci after 3 Gy of photon and proton irradiation, it was reported that photon irradiation resulted in the formation of small γH2AX foci that were equally distributed within the nucleus. Proton irradiation in the SOBP composed of 6 Bragg peaks of 100–110 MeV, on the other hand, induced the formation of larger γH2AX foci that were more heterogeneously distributed within the nucleus [67]. Multiple other studies have also observed larger γH2AX foci after proton compared to photon irradiation, which could be due to the clustered nature of the DNA damage induced by proton radiation [68,69,70,71,72,73]. In addition, slower repair kinetics after proton therapy have been observed [30,67,71,73,74,75,76]. This is indicative of clustered DNA damage, which is thought to result in slower repair [77].

Whether proton irradiation causes more DNA damage than photon irradiation was also investigated by immunofluorescence detection of γH2AX foci. It was found that the number of γH2AX foci induced after 0.5, 1, and 2 Gy of 200 MeV proton radiation compared to 10 MV photon irradiation were significantly higher in both medulloblastoma and leukemia cells [72]. Similarly, using pulse-field gel electrophoresis, the number of DSBs induced by 25 Gy of 76 and 201 MeV proton irradiation was higher compared to similar doses of photon irradiation in head and neck cancer cell line SQ20B. This increase in DSBs was seen for samples irradiated with entrance, mid-SOBP, and distal-SOBP protons [27]. Additionally, a comet assay analysis revealed a higher number of both SSBs and DSBs after 10 Gy proton irradiation compared to 10 Gy 320 kV photon irradiation in glioblastoma stem-like cells [57]. However, the number of initial γH2AX foci was the same in Chinese hamster ovary cell lines treated with 138 MeV SOBP proton or 200 kV photon irradiation of 1 Gy [78]. In addition, in prostate cancer cells and murine embryonic fibroblasts irradiated with 3 Gy 187 MeV Bragg-peak or plateau protons compared to 320 kV photons, no difference in initial γH2AX foci could be observed. However, the maximum number of foci was reached 60 min after irradiation with Bragg-peak protons contrary to 30 min after irradiation with plateau protons and photons [71].

Overall, in vitro research seems to indicate that more DNA damage is induced after proton radiation that could be of a clustered nature as indicated by Monte Carlo simulations.

### 4.2. Signaling of Clustered DNA Damage after Proton Irradiation

Together with DSBs, clustered DNA damage is considered as the major contributor to cell killing by radiation. Due to the clustered nature, cells experience difficulty repairing these lesions, which leads to higher residual damage in cells [79]. Because clustered DNA damage consists of different DNA lesions, multiple pathways may be involved in the repair of these lesions, like, e.g., a combination of homologous recombination (HR) or nonhomologous end-joining (NHEJ) and base excision repair (BER) [80,81]. Consequently, induction of clustered DNA damage plays a critical role in the effect of proton radiation. However, the cellular response to these types of lesions is not very well understood. Similar to H2AX phosphorylation to signal the presence of a DSB, there might be a signaling mechanism to activate the repair of clustered DNA damage. Indeed, recently, it was reported that ubiquitylation of lysine 120 on histone H2B is promoted in HeLa and head and neck squamous cell carcinoma (HNSCC) cells after irradiation with 11 MeV protons. Ubiquitylation is the post-translational process of attaching a small protein called ubiquitin to another protein. When DNA damage occurs, this process is part of the regulation of the DNA damage response, as reviewed in [82,83,84]. The ubiquitylation of histone H2B was catalyzed by E3 ubiquitin ligases ring finger 20/40 clustered (RNF20/40) and male-specific lethal 2 homolog (MSL2). Using a neutral comet assay modified to detect and measure DSBs and oxidative clustered DNA lesions they found a linear dose response for the induction of these clustered lesions [85]. The enzyme-modified neutral comet assay for siRNA depleted RNF20/40 and MSL2 cells revealed more clustered DNA damage after irradiation compared to control cells. As a result, this ubiquitination event was essential for the efficient repair of clustered DNA damage [76]. In a follow-up study of the same group, a siRNA screening in HeLa and HNSCC cells for deubiquitylation enzymes involved in the promotion of cell survival after 58 MeV protons compared to 100 kV photons was performed. They showed that ubiquitin-specific protease 6 (USP6) stabilizes the protein poly(ADP-ribose) polymerase 1 (PARP-1), leading to higher cell survival. As PARP-1 is involved in SSB repair, it can be used as a marker for SSBs. In cells depleted of USP6, higher levels of clustered DNA damage were found using the enzyme-modified neutral comet assay, which led to a cell cycle arrest and a deficient repair of clustered DNA damage [86]. Taken together, they propose the following model for the cellular response to clustered DNA damage induced by proton irradiation: clustered DNA damage triggers ubiquitylation of H2B mediated by RNF20/40 and MSL2. This allows the recruitment of DNA repair proteins and chromatin remodeling factors to promote the accessibility of the clustered lesion. PARP-1, stabilized by USP6, is thought to play a critical role in efficient repair of clustered DNA damage [77].

### 4.3. Differential Repair of Photon- and Proton-Induced Lesions

When DNA damage is induced, DNA damage response pathways are activated in order to start the correct repair mechanism. From the DNA damage that is induced by ionizing radiation, double strand breaks (DSBs) are considered the most lethal. These DSBs are repaired by two major pathways: homologous recombination (HR) and nonhomologous end-joining (NHEJ) [87,88]. HR uses a homologous template strand as a template to repair the DSB. As a result, HR is mainly active during the S and G2/M phase of the cell cycle when the chromosomes are duplicated and a homologous template is present [89]. The use of this homologous template results in preserved sequence integrity after HR-mediated repair of DSBs. However, mammalian cells predominantly use the faster NHEJ repair pathway to deal with DSBs, which is a more error prone pathway active throughout the whole cell cycle.

Due to their complex nature, clustered DNA damage is thought to be preferentially repaired by HR [77]. Several papers have indeed shown that cells with an impaired HR machinery are more sensitive to proton irradiation [67,78,90,91,92,93]. Grosse et al. [78] showed that Chinese hamster ovary (CHO) cells deficient in HR and NHEJ were more sensitive towards 200 kV photon as well as 138 MeV SOBP proton radiation compared to wild-type cells (without deficiencies in HR and NHEJ). However, only cells deficient in HR showed hypersensitivity towards proton therapy, which resulted in an increased RBE compared to the RBE calculated for the wild-type (1.44 and 1.29, respectively, at 10% survival fraction) [78]. Similarly, fibroblasts and cancer cells with deficient NHEJ were sensitive to both photon and proton irradiation, but fibroblasts with a deficient HR showed enhanced radiosensitivity to only proton irradiation compared to wild-type fibroblasts. This effect was seen when cells were irradiated with entrance plateau protons and became even bigger for SOBP protons when the same physical doses were tested for all irradiations with a calculated RBE of 0.98 and 1.12, respectively [67]. This effect could be due to the increase in LET along the Bragg peak profile, which, in turn, results in an increased induction of clustered damage. Research conducted in yeast also showed that HR-deficient yeast strains are sensitive to 250 MeV proton radiation [93]. However, it should be noted that yeast mainly uses HR as a repair pathway for DSBs and does not rely as much on NHEJ as compared to mammalian cells.

The dependency on HR for the repair of these clustered lesions implies that inhibiting NHEJ might sensitize cells more to photon than to proton therapy. In fact, Fontana et al. [90] observed a radiosensitizing effect of a DNA-PKcs inhibitor for 200 kV X-rays but not as much for proton radiation (SOBP, 138 MeV). This effect was correlated with a delay in repair kinetics after photon radiation as examined through γH2AX immunofluorescent staining after 1 Gy of photon or proton irradiation. In addition, they observed fewer phospho-DNA-PKcs foci in proton-irradiated cells, compared to photon irradiation, which again suggests differential requirement of HR and NHEJ in response to photon or proton radiation [90]. A similar study in glioblastoma cells showed that HR-deficient cells are sensitive to protons, but less to photons. However, they highlight that NHEJ remains the most important repair pathway for radiation-induced damage [94]. Generally, these results seem to show that NHEJ remains crucial for the repair of DSB irrespective of the type of irradiation, however, HR becomes more important when cells are irradiated with high LET radiation.

In contrast to the results discussed above and to what is generally accepted, some groups have also found no greater dependence on HR after proton compared to photon radiation. One study observed no significant differences in γH2AX foci formation and repair in wild-type and NHEJ-deficient Chinese hamster ovary cells after photon and proton irradiation. Proton irradiation was performed at the center of a SOBP with an energy range of 0–60 MeV. However, they did observe an increased involvement of HR in the repair of DSBs for carbon ion beams (SOBP, 0–160 MeV). This effect was confirmed by the RBE values calculated at 10% survival. For protons, this RBE value ranged from 0.89 to 1.10. No differences were found for the wild-type, HR-deficient, and NHEJ-deficient cells. However, for carbon ions, the RBE ranged from 1.07 to 2.10. The RBE values for the wild-type and HR-deficient cells were significantly higher than those of the NHEJ-deficient cell lines [95]. Another group investigated the repair kinetics in wild-type and NHEJ deficient cell lines after irradiation with photons and protons. Similarly, they found no differences in biological effects for photon or entrance and Bragg peak proton irradiation [73]. It should also be noted that the observed effect can be dependent on multiple factors including tested doses, observed RBE values, position of irradiation in the Bragg peak profile, proton and photon energies that are used, as well as cell lines that are tested. In the studies mentioned above, a wide variety of these depending factors is used, which might partially explain the heterogeneous results. This highlights the need for more, well-controlled studies to be able to give a definite answer to the involvement of HR versus NHEJ to repair proton-induced DNA damage. Experiments with the same photon and proton energies should be repeated in multiple cell lines to check if the repair pathway choice depends on the type of cell line. Within one cell line, different proton energies and irradiation positions can then be tested.

### 4.4. Cell Cycle Progression after Photon and Proton Irradiation

When cells experience DNA damage, DNA damage checkpoints are activated and control the progression through the cell cycle. The repair pathway that is activated is also dependent on the cell cycle stage. Consequently, the cell cycle progression of cells irradiated with photon or proton therapy is another important research topic. Up until now, no consistent results have been reported about the cell cycle distribution after photon compared to proton therapy. Like for photon therapy, a G2/M arrest was observed in human lung cancer cells irradiated with multiple doses of 62 MeV protons [96]. In addition, in PC3 and MCF7 cell lines irradiated with 10 Gy 26.7 MeV protons a G2/M arrest was found [55]. However, in both cases, a direct comparison with photon therapy was missing. Other studies also reported an accumulation in G2/M for HeLa, glioblastoma cells, and fibroblasts using 58, 5.7, and 0.8 MeV protons, respectively [76,86,97,98]. For the glioblastoma cells and fibroblasts, this arrest was more pronounced compared to 120 and 250 kV X-rays, respectively [97,98]. In addition, in non-small cell lung cancer cells (NSCLC) irradiated with protons, a higher and prolonged G2/M arrest was found compared to photon irradiation for equitoxic doses as an RBE of approximately 2 was calculated [34]. On the other hand, another research group found a shorter G2/M arrest in glioblastoma stem-like cells irradiated with protons compared to 320 kV X-rays for multiple tested doses. A similar cell cycle distribution was observed 3 days after irradiation, however 6 days after irradiation, cells treated with proton irradiation were recovered from the G2/M arrest while this recovery was not observed after photon irradiation. They found that this was the result of sustained induction of CHK2 phosphorylation and more rapid CHK1 dephosphorylation after proton radiation. CHK1 and CHK2 are two proteins that function as regulators of the cell cycle. They conclude that proton irradiation may induce stronger CHK2 phosphorylation to reduce the recovery time for the G2/M arrest [57]. Hartfiel et al. [99] assessed the cell cycle distribution after 2 and 8 Gy of photon and proton radiation at the SOBP in four esophageal cancer cell lines. They found a G2/M arrest after both photon and proton irradiation for all cell lines, however, reported a prolonged G2/M arrest after proton irradiation for only two out of four tested cell lines [99]. In Chinese hamster ovary cells, no differences were found in the cell cycle progression after irradiation with 5 Gy 138 MeV SOBP protons or 200 keV X-rays [78]. As with the DNA repair experiments, multiple photon and proton energies as well as positions in the SOBP profile were used when conducting these experiments.

Even though there are no consistent results regarding the cell cycle distribution after proton irradiation compared to photon irradiation, studies have also shown differential gene and protein expression patterns for cell cycle-related genes and proteins after photon and proton irradiation. An in vivo study, where mice were irradiated with either 7 Gy 1 GeV plateau protons or 6.4 Gy γ-rays showed that the expression of genes in the categories of apoptosis, cell cycle regulation, and the DNA damage response were most significantly different. For the cell cycle-related genes, they reported that protons induced more cell cycle stimulators, while photons induced more cell cycle blockers [100]. More recently, a proteomic analysis in head and neck cancer cells revealed differential protein expression patterns after irradiation with 4 Gy 6 MV photons and 200 MeV SOBP protons. Contrary to the in vivo study, they reported a higher expression of proteins involved in the DNA damage response and cell cycle arrest after proton radiation, while photon radiation induced a higher expression of cell cycle progression-related proteins [101].

## 5. Cell Death after Proton Irradiation

Cell death induced by radiation is an important part of the effect of a radiation treatment. The main types of radiation-induced cell death are mitotic death, apoptosis, necrosis, senescence, and autophagy [102,103]. In mitotic catastrophe, cells are unable to complete their mitotic division due to an early entrance in mitosis, while the DNA is still unrepaired. As a result, a mitotic arrest will be triggered, which eventually leads to cell death [104]. Apoptosis is a highly regulated cell death process, where a controlled degradation of cellular components is activated [105,106]. Necrosis on the other hand is an uncontrolled type of cell death provoked by external factors. Senescent cells are in a permanent cell cycle arrest. This senescent state is triggered by radiation-induced DNA damage and the induction of p53 and pRb pathways that lead to a cell cycle block [104]. Lastly, autophagy is the well-regulated mechanism of the cell to remove damaged or old cytoplasmic organelles in response to stress [107]. One of the main factors determining the type of cell death after irradiation is the cell type. For hematopoietic cancers, it is known that radiation treatment results mainly in apoptosis [108]. However, for most solid tumors, mitotic catastrophe is the most important cell death type [109].

Besides differences in types of death depending on the cell type, photon and proton radiation might also influence cell death-related processes in the cell in a differential manner. There appears to be an agreement in literature that proton radiation causes more apoptosis than photon irradiation [55,56,57,58,72,100,110]. Moreover, this effect seems to be caused by an increase in ROS production [56,57,58]. However, fewer research has been conducted into the different modes of cell death after photon compared to proton irradiation. Miszczyk et al. [111] investigated the rate of cell killing by apoptosis or necrosis in an ex vivo human peripheral blood lymphocyte model after irradiation with 250 kV photons or 60 MeV SOBP protons. They found that irradiation with protons caused more cell death compared to photons and that the mode of cell death was mostly necrosis [111]. Wang et al. assessed cell death in 4 HNSCC cell lines after a single dose of 4 Gy 6 MV photons or 200 MeV SOBP protons. No differences were found in the level of necrosis and apoptosis. However, more senescence and mitotic catastrophe were observed after proton irradiation [112]. The difference in the results can largely be attributed to the different cell types. The main cell death pathway for blood lymphocytes is known to be apoptosis, while for HNSCC cells, the main pathway is mitotic cell death. Moreover, Miszczyk et al. [111] did not investigate the levels of senescence and mitotic catastrophe.

Apart from the differences in the amount of cell death and the used mode of cell death, there also seems to be a difference in signaling mechanisms between photon and proton irradiation to induce apoptosis. Murine macrophages were exposed to 2 Gy photon or proton radiation. Activation of the proapoptotic p38 pathway showed a gradual increase over time after proton irradiation. Photon irradiation resulted in a decrease in phosphorylated p38, which returned to control levels 2 h after the irradiation. After proton irradiation, marginal activation of the prosurvival extracellular signal-regulated kinase (ERK) pathway was observed at all timepoints investigated. After photon irradiation on the other hand, activation of ERK occurred in a peak at 2 h after irradiation [113]. Similarly, increased mRNA expression of proapoptotic genes in human tumor cell lines PC-3, MCF-7, and Ca301D was observed after irradiation with 10 Gy of 26.7 MeV protons that was several fold higher compared to irradiation with 10 Gy of 15 MV photons [55]. The cell death signaling mechanisms seem to differ between photon and proton radiation, however, the relation between the signaling and the complexity of the induced DNA damage has not been clarified. A different regulation of cell death can implicate a different response to certain chemotherapeutic drugs. Consequently, a direct translation of drugs used in combination with photon radiotherapy might not be possible. Therefore, it is important to know more about the regulation of cell death after proton compared to photon irradiation.

## 6. Proton Therapy to Overcome Radioresistance

Radioresistance and tumor recurrence still pose a major problem in radiation treatment. To eradicate these radioresistant tumors, doses higher than usual are needed. However, in clinical practice, this dose escalation is not always possible due to toxicity to the normal tissue. Previous studies have identified mechanisms enabling radioresistance to conventional photon radiotherapy among which are hypoxia, cancer stem cells (CSCs), and mutations in survival pathways and DNA damage repair pathways [114,115]. The potential of proton irradiation to overcome this therapeutic radioresistance is still understudied, however, an overview will be given about the available literature.

We already discussed how protons have additional benefits in overcoming hypoxia-mediated radioresistance compared to photon radiation. However, additionally, protons seem to be more efficient in eradicating cancer stem cells (CSCs). CSC are intrinsic radioresistant cells within a tumor that possess the capacity to self-renew and to cause the heterogeneous lineages of cancer cells that comprise the tumor [116]. The radioresistance of CSCs can be attributed to differences in DNA-repair capacity, their quiescent state and hypoxic tumor microenvironment as well as an enhanced ROS defense in response to conventional radiotherapy [117]. An in vitro study showed that CSC-enriched non-small cell lung cancer cells were more sensitive to proton irradiation compared to photon irradiation. They hypothesized that this increased sensitivity towards proton irradiation could be caused by the higher levels of ROS generated by protons, which, in turn, caused more apoptosis [58]. Similarly, in glioma stem cells higher ROS levels after proton irradiation led to reduced clonogenic survival compared to photon radiation [57]. Additionally, proton irradiation was shown to be more efficient to kill CSCs at the same dose in breast cancer cell lines [118]. Similarly, in NSCLC, proton radiation was more effective in reducing the population of CSC-like cells compared to photon radiation [34]. All these results seem to indicate that proton therapy is better suited to kill CSCs and, as such, could be used to treat radioresistant tumors.

Signaling pathways and the way they are influenced by radiation can also contribute to radioresistance. For example, signaling of DNA damage is an important part of the response to radiation. In this context, PARP functions in the repair of SSBs as well as DSBs and is thus an important factor in DNA damage signaling. Inhibition of PARP sensitized lung cancer and pancreatic cancer cell lines to 160 MeV protons at both the entrance region and Bragg peak [119]. PARP also plays a role in apoptosis, as cleaved PARP levels are an indicator of apoptosis. Higher and more sustained cleaved PARP levels were reported after proton irradiation [57]. Activation of survival pathways also plays a role in radioresistance leading to treatment failure. Recently, in HNSCC cell lines, it was found that 4 Gy 6 MV photon irradiation induced more procell proliferation and survival proteins, while 4 Gy 200 MeV SOBP proton therapy caused the expression of more anticell proliferation and growth proteins [101]. This could again be a potential benefit of proton radiation in overcoming radioresistance.

## 7. Conclusions

Clinical use of protons has massively improved since its first applications in 1954. However, radiobiological research has been hampered by the scarcity of accessible proton facilities for research. Early results pointed towards differential proton and photon radiobiology, however, the exact mechanisms remain largely understudied. Knowing about the differential radiobiology of proton radiation compared to photon radiation can become important to determine optimal therapeutic strategies. This is especially relevant for personalized medicine. Understanding the radiobiology of protons will allow to integrate clinical, physical, and biological parameters to adjust treatment to the specific case of an individual patient. Together with the immense increase in proton therapy facilities worldwide in the last 10 years, integrated research facilities are becoming more available. As a result, our knowledge about proton radiobiology is expanding. In this review, an overview was given about the current knowledge of proton radiobiology. A summary can be found in Table 1.

From Monte Carlo simulations, it is expected that irradiation with a higher LET causes more clustered DNA damage. Multiple studies have reported larger γH2AX foci and slower repair kinetics after proton radiation, which can indicate the induction of more clustered DNA damage as this type of DNA damage is preferentially repaired by HR. Therefore, the relative dependence on HR or NHEJ to repair DNA damage induced by protons has also been investigated. However, inconsistent results have been reported. Generally, HR seems to become more important for the repair of DNA damage induced by higher LET radiation, which is in agreement with the common believe that clustered DNA damage is preferentially repaired by HR. However, NHEJ remains indispensable in the repair of radiation-induced DNA damage. There is experimental evidence that proton radiation, similar to photon radiation, induces a G2/M arrest. However, regarding the extent and duration of this G2/M arrest compared to photon radiation, conflicting results have been reported. Besides differences in the DNA damage response, protons have additional differential effects on other processes in the cells compared to photons. It has been reported that proton radiation induces more ROS compared to photon radiation. Additionally, protons cause more extensive cell death. Moreover, a differential regulation of cell death has been observed where protons induce more proapoptotic signals. Besides, protons have the potential to be more effective in the treatment of radioresistant tumors.

More and more evidence is arising that photon and proton radiobiology differ in more aspects than previously expected. However, the heterogeneity of cell lines and different photon and proton irradiation modalities, such as the dose, dose rates, energies, and positions along the Bragg peak profile, complicate the comparison of the results. Therefore, it is important to check whether results can be reproduced with different proton energies, and thus different LETs, in order to get conclusive answers. Further research to understand the mechanisms that differ after photon and proton irradiation is important for their possible translation to the clinic.

## Figures and Tables

**Figure 1 cancers-13-00604-f001:**
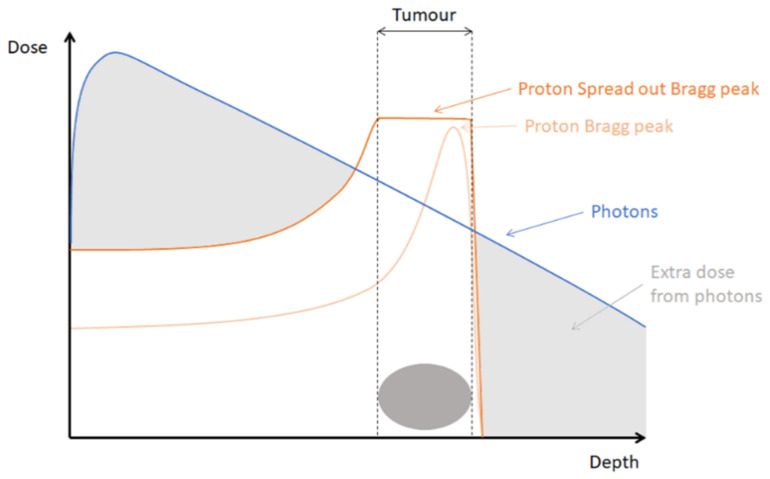
Depth-dose distribution of photons compared to protons.

**Table 1 cancers-13-00604-t001:** Summary of the radiobiology of proton radiation compared to photon radiation.

Biological Process	Photon Irradiation	Proton Irradiation
Relative biological effectiveness		General RBE of 1.1 is assumed in planning Biological data reveal varying RBE
DNA damage response	NHEJ is main repair pathway	Induction of clustered DNA damage leading to larger γH2AX foci and slower repair kinetics HR becomes more important for repair
Cell cycle distribution	G2/M arrest	G2/M arrest can be more extensive and prolonged
ROS		Induction of more ROS
Cell death	Activation of prosurvival response	Induction of more cell death Activation of proapoptotic response
Radioresistance	Hypoxia: increased expression of HIF-1α CSCs are less sensitive Activation of prosurvival pathways	Hypoxia: decreased expression of HIF-1α CSCs are more sensitive Decreased activation of prosurvival pathways

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
