# Peer review of "Clinical Progress in Proton Radiotherapy: Biological Unknowns"

_cancers, 2021, doi:10.3390/cancers13040604_

Round 1

Reviewer 1 Report

General Comments:

The advent of proton therapy in clinical treatments necessitates an update of out knowledge of the effects of proton radiation on living materials. The authors of this review provide a timely overview of the literature in a broad approach of all subjects impacting on said effects.

They include RBE effects, Oxygenation effects, DNA damage response , cell Death, and radiation resistance. The gathered articles are relevant and provide a good overview of the current knowledge in the field.

I have reviewed this paper as a general review which needs to be read by researchers with a varied background. Some of the statements therefore can sound somewhat naive.

Unfortunately, this work lacks a critical voice in the review of the papers, nor is there an overarching theme in the treatment of the results, but rather a summation of the results of the paper. In addition a number of statements are not supported or are considered to be received knowledge without critical assessment, making this paper not acceptable at this point in time. The review also feels like too much ground is covered and sometimes feels like there are different research approaches which are competing. Indeed, sometimes it reads like a physics paper, a clinical paper, a microbiology paper and a molecular biology paper all with their own jargon.

There are two possible paths the authors can take. Perform an extensive rewrite, which would probably require a large effort and should be done by a senior person. Alternatively, a more focused approach regarding a subset of the current paper. For instance cell death or radiation resistance using an underlying model as a guide (see specific comment).

Specific comments:

A recurrent theme is the increase in clusters of complex damage which in some experiments occur and in others not. This seems to be driving the difference in biological response (reading between the lines). The authors should be very clear in how complex damage is defined So what is the difference between Double strand breaks, clusters of complex damage etc…

Another overall comment is the cavalier attitude to the specific parameters under which all reported experiments with protons are taken. This is a general problem in the current radio biology research, and due to a historic reliance on photon radiation as an experimental tool . Indeed, photon based radiation experiments are equivalent to subjugating the samples to a spectrum of electrons, which is almost invariant with respect to position within the beam, this is combined with the very flat response of DNA damage with respect to the electron energy. Such that the source 250kVp to 20MV photons give roughly the same results and it thus a very robust tool. When using protons things are quite different.

For one protons are modulated (their kinetic energy reduces depending on the penetration depth). This implies that a different spectrum of protons impacts on the cells depending in where in the beam they are placed. In addition the difference in energy generates different LET and ionisation densities.,

sometimes densities comparable with electrons some times much higher. Making this a much less reliable experimental source and one in which care needs to be taken to describe the exact conditions.

This should be acknowledged throughout the paper and make it clear when some results can be compared and when not.

Detailed comments:

Line 46: Replace : Modifying the energy to accelerate the protons, By: Modifying the energy of the accelerated the protons, this will include protons moderated by physical means. Indeed most cyclotrons operate on a single energy and subsequently these are moderated by a wedge type attenuator, synchrotron based proton machines are able to switch energy within the accelerator but they form a minority in the proton therapy landscape).

Line 55: Use high instead of great.

Line 97: The definition of LET is quite pedestrian here; The quantity as defined in the text : “the amount of energy deposited per unit distance” is called collisional Stopping Power. LET narrows that to energy reduced by the energy taken away by secondary electrons which can travel substantially from the primary charged particle’s track. This reduces the energy transfer concept to occur along the track only and connects is nicely to the concept of ionisation density along a line.

Line 123: Reading between the lines, the authors attribute the oxygen effect to indirect damage as that generates reactive oxygen species. It seems to me that there is a confusion. The reactive oxygen species are generated also when there is no oxygen available as they are a by-product of the radiolysis of water. The authors do not mention another concept which is the oxygen fixation mechanism and the competition with chemical repair which is likely to be the driving mechanism. As such hypoxia does not lead to radiation resistance it is the oxygen which acts as an enhancer. The OER is likely also dependent on where in the SOBP the assessement is made.

Line 126: What is considered to be high LET? We know that there is a notion that if the LET increases that then something different happens, but what would be a threshold? As used here it seems to be quite arbitrary.

Line 134: The paragraphs with respect to HIF1-α expression are not clear. The only example provided is the exposure to 1GeV protons (an energy that is not clinically relevant) and which likely generates a field of additional particles (neutrons, recoil heavy ions) which will contribute to any effect. There is no indication or mechanism mentioned as to why the expression changes when protons are used. Is this because of increased complex damage which inhibits the expression? As mentioned above it is not clear what is meant by an experiment with 1GeV protons, do the protons that interact with the sample have an energy of 1GeV, is this kinetic energy, total energy? Or is the sample placed at the bragg peak where the energy is much lower and comparable to any other bragg peak.

Line 142: The increase in ROS of protons with respect to photons is not something that has a relationship to oxygenation but should be mentioned in the RBE section. It is likely that this is related to an increase in ionisation density, which induces more ionisations and ROS generation close to the DNA structure. Generating not only more direct damage but also a higher probability for indirect damage.

Line 153: The final line does not fit here as we have not mentioned the impact the cell cycle yet. It is not clear how the previous explanation leads here. How does the ROS scavenger (also not defined ) affect the start of a cell cycle transition?

Line 159: The sentence: “ increasing LET causes more extensive and more clustered DNA damage as a result of denser ionization events”. Might be clearer as follows:

increasing LET results in denser ionisation events, which causes more extensive and more clustered DNA damage

Line 161: This discussion might be better placed in the section on RBE or vice versa.

Line 163 : How does the presence of repair enzymes make the detection of DNA damage difficult? I think I know what the authors are thinking of (i.e. the damage is repaired and therefore can’t be detected). However, the relevant damage the will lead to cell death is not repaired and should therefore still be detectable. Also one needs to be aware that the enzymatic repair process happens on a vastly different time scale compared to the molecular damage processes. But this is under the assumption that I can read minds, so the authors need to be more specific.

Line 175: A lot of jargon starts to be used here, please define terms like histone, phosphorylation

(I know what you mean but this is an overview which should provide information to different types of specialties in research).

Line 180: Again there is the issue that the authors are not specific in what exactly is meant by proton radiation.

Line 190 -203: It is not clear whether γH2AX differentiates between DSB and complex cluster damage . The distinction is not clear and the terms are used in a confusing manner. Again not clear how the plateau was achieved and what the spectrum was there. One could make the statement that a the bragg peak we would get more complex damage as ionisation density would be higher.

Line 205: A differentiation of Double strand Breaks and complex cluster damage is made. The difference between these two is not clear.. It should be made clear how this is being determined.

In addition there is an indication that for repair purposes there is a difference between them. As indicated in the text. It is not clear how the different pathways are altered depending on the presence of clustered damage they are just mentioned.

Line 213: Suddenly, the language in this review changes and it is only suited for a reader well versed in molecular biology. While earlier physical terms were somewhat defined, This has been thrown overboard. Many of your readers will not know what ubiquitylation is

This ubiquitylation was catalyzed by E3 ubiquitin ligases ring finger 20/40 clustered (RNF20/40) and male-specific lethal 2 homolog (MSL2)” . This is completely unreadable, but for the initiated few.

Line 127: Using a neutral comet assay modified to detect and measure DSBs and oxidative clustered DNA lesions they found a linear dose response for the induction of these clustered lesions

Again not clear how the comet assay is linked to the previous part. Would one not expect an increase of lesions if the dose is increased. I fail to see the link with the previous part. Does a comet assay detect whatever you talked about. It now seems that you are able to detect DSB, as well as clustered damage and other things that was previously indicated as impossible. Further even you now say that SSB’s are repaired by PARP although we are led to believe that cell death is governed by DSB and complex damage. Again a process which is suddenly mentioned.

Line 228: This part shines some light on the previous darkness. So it might be interesting to put this upfront such that you illustrate the model which is being developed and support it with data. Also this is the good place to introduce in more simple terms what the ubiquitilation process entails.

Line 233: A good place to force a new page

Line 234: This could be a more philosophical remark. While the previous part of the text has involved the generation of damage and complex processes which the DNA-molecules and cell undergo as a passive bystander . The language now changes to an active proponent where the cell becomes an active subject “it relies on signaling”? You make it seem that the cell actively decides to signal and activates the repair mechanisms. This is an antropomorphism, which unfortunately is rife in the radio-biology community. You imply that there is an intellectual process were the damage is identified and subsequently a repair pathway is chosen (on purpose) At this time there is no need for this and the remainder of this section can be well read without resorting to this.

Line 238: This is a good place to introduce the enzymatic repair mechanisms. Specifically, those that use templates. This because as you fail to mention these templates become available when more complex damage is being introduced which could be seen as a driver of which repair mechanisms prevail. Line 244-277 then becomes more plausible.

Line 248: Define “wild type” cells (I know what they are but others do not).

Line 255: Can this be related to increased complexity? In the previous sections you have established that the difference between photons and protons is increased complexity in damage and the complexity increases downstream of a SOBP. Rather than doing that we obtain a summation of results. See also Line 277.

Line 282: A remark is provided that the results outlined above depend on several factors and that these are not disclosed adequately. For instance looking at the Carbon ion results with RBE varying from 1.07 to 2 , which leads me to presume that high energy carbon ions were used in the first. So if you propose to have controlled studies you might want to highlight what needs to be controlled and whether there could be a common language or model.

Line 296: This sentence is difficult to read. Results are presented from the HeLa-, glioblastoma-cells and fibroblasts. The next sentence then only talks about glioblastoma-cells and fibroblasts as comparison. So either the HeLa cells were not compared or did they not show a difference. If it is the latter then the energy of the protons is important (as are the experimental circumstances) as 48MeV protons have a ionisation density comparable with photons.

Line 304:

They found that this was the result of sustained induction of CHK2 phosphorylation and more rapid CHK1 dephosphorylation after proton radiation [57]”

None of this was previously defined and not for the uninitiated.

Line 313: Could you define what you mean by “expression”? Again I think I know what this but it would be great if a definition is provided. Also suddenly you speak of expression of genes after radiation, as genes of cells are altered through evolutionary processes I would not expect that they would be affected in these short terms, but rather that they are selected out? It is also the first time genes are mentioned in this paper, so a short sentence of the importance in this context would be nice.

Line 340: There is an apparent contradiction between the data presented by Miszczyk et al compared to Wang et al. This should be commented on. Different cells and different proton energies?

Line 356: Activation of pathways is now introduced. While they are well studied in radio—biology the link with differences of complexity of damage is not clear. So in this context more information on the mechanism and its relationship to damage is interesting

Line 386: Rather than hypoxia making a cell radio resistant it seems to be the other way round that oxygen makes cells more sensitive and enhances the damage.

Line 394: Quite a late stage to define PARP.

Line 415: Usually this type of Monte Carlo is different than general purpose MC codes, This is usually considered microdosimetric monte Carlo simulations, examples are PARTRAC, MCDS, LEM IV. Also noteworthy that as the LET increases not only the ionisation density changes but also the geometry. In low LET the foci are distributed randomly through the cell nucleus, while at very large LET the follow the track line closely.

Author Response

Review report reviewer 1

General comments:

The advent of proton therapy in clinical treatments necessitates an update of out knowledge of the effects of proton radiation on living materials. The authors of this review provide a timely overview of the literature in a broad approach of all subjects impacting on said effects.

They include RBE effects, Oxygenation effects, DNA damage response , cell Death, and radiation resistance. The gathered articles are relevant and provide a good overview of the current knowledge in the field.

I have reviewed this paper as a general review which needs to be read by researchers with a varied background. Some of the statements therefore can sound somewhat naive.

Unfortunately, this work lacks a critical voice in the review of the papers, nor is there an overarching theme in the treatment of the results, but rather a summation of the results of the paper. In addition a number of statements are not supported or are considered to be received knowledge without critical assessment, making this paper not acceptable at this point in time. The review also feels like too much ground is covered and sometimes feels like there are different research approaches which are competing. Indeed, sometimes it reads like a physics paper, a clinical paper, a microbiology paper and a molecular biology paper all with their own jargon.

There are two possible paths the authors can take. Perform an extensive rewrite, which would probably require a large effort and should be done by a senior person. Alternatively, a more focused approach regarding a subset of the current paper. For instance cell death or radiation resistance using an underlying model as a guide (see specific comment).

Since both reviewers 2 and 3 agreed with the paper without any comments, we decided not to perform an extensive revision of the paper since this would be against the advice of both reviewer 2 and 3 . The aim of our paper was to give a broad overview of the current biological knowledge on proton radiotherapy, and more specifically the gaps that are still missing.

Specific comments:

A recurrent theme is the increase in clusters of complex damage which in some experiments occur and in others not. This seems to be driving the difference in biological response (reading between the lines). The authors should be very clear in how complex damage is defined So what is the difference between Double strand breaks, clusters of complex damage etc…

We have tried to make the definition of clustered DNA damage more clear by giving examples of what types of DNA damage a clustered DNA damage site could be composed of (see line 165-171).

Another overall comment is the cavalier attitude to the specific parameters under which all reported experiments with protons are taken. This is a general problem in the current radio biology research, and due to a historic reliance on photon radiation as an experimental tool . Indeed, photon based radiation experiments are equivalent to subjugating the samples to a spectrum of electrons, which is almost invariant with respect to position within the beam, this is combined with the very flat response of DNA damage with respect to the electron energy. Such that the source 250kVp to 20MV photons give roughly the same results and it thus a very robust tool. When using protons things are quite different. For one protons are modulated (their kinetic energy reduces depending on the penetration depth). This implies that a different spectrum of protons impacts on the cells depending in where in the beam they are placed. In addition the difference in energy generates different LET and ionisation densities. sometimes densities comparable with electrons some times much higher. Making this a much less reliable experimental source and one in which care needs to be taken to describe the exact conditions. This should be acknowledged throughout the paper and make it clear when some results can be compared and when not.

We agree with the reviewer that the conditions under which all experiments were performed in all the referred papers are not homogenous. This is however an intrinsic issue in biology research as such. We can only report the parameters under which the experiments were performed. To make the experiments more clear we have added the position of irradiation in the text when reported in the referred paper (line 195-196, 208-209, 212, 272-274, 280, 292-293, 294, 300-301, 302-303, 333, 337, 343, 348, 374, 376, 428-429, 432, 433).

Detailed comments:

Line 46: Replace : Modifying the energy to accelerate the protons, By: Modifying the energy of the accelerated the protons, this will include protons moderated by physical means. Indeed most cyclotrons operate on a single energy and subsequently these are moderated by a wedge type attenuator, synchrotron based proton machines are able to switch energy within the accelerator but they form a minority in the proton therapy landscape).

Thank you for the input. We have changed the phrasing in line 46.

Line 55: Use high instead of great.

We followed the suggestion and changed ‘high’ to ‘great’ in line 55.

Line 97: The definition of LET is quite pedestrian here; The quantity as defined in the text : “the amount of energy deposited per unit distance” is called collisional Stopping Power. LET narrows that to energy reduced by the energy taken away by secondary electrons which can travel substantially from the primary charged particle’s track. This reduces the energy transfer concept to occur along the track only and connects is nicely to the concept of ionisation density along a line.

We added the words ‘along the particles track’ in line 98 to the definition of LET to the differentiate between LET and Stopping power.

Line 123: Reading between the lines, the authors attribute the oxygen effect to indirect damage as that generates reactive oxygen species. It seems to me that there is a confusion. The reactive oxygen species are generated also when there is no oxygen available as they are a by-product of the radiolysis of water. The authors do not mention another concept which is the oxygen fixation mechanism and the competition with chemical repair which is likely to be the driving mechanism. As such hypoxia does not lead to radiation resistance it is the oxygen which acts as an enhancer. The OER is likely also dependent on where in the SOBP the assessement is made.

The reviewer is indeed correct about the role of oxygen. It is indeed well known that local availability of molecular oxygen enhances the efficacy of radiotherapy, as DNA lesions caused by ROS produced during water radiolysis react with oxygen to form stable DNA peroxides. Thus cancer cells that receive less or consume more oxygen are typically more radioresistant than well oxygenated cancer cells. As written in lines 125-127 ‘Consequently, cells are more sensitive to radiation when oxygen is present and as a result tumor hypoxia can lead to radioresistance [43,44].’ we state that oxygen is the crucial enhancer. In line 108 we also stated that there is increasing evidence that the proton RBE varies along the SOBP, and this is partly explainable by variations in oxygen levels. We have clarified this by adding ‘by reacting with molecular oxygen to form stable DNA peroxides’ in line 122.

Line 126: What is considered to be high LET? We know that there is a notion that if the LET increases that then something different happens, but what would be a threshold? As used here it seems to be quite arbitrary.

We agree with the reviewer that the cut-off high versus low LET is quite arbitrary without a fixed threshold. In literature, high-LET radiation are considered protons or neutrons in contrast to low-LET radiation (gamma rays). As mentioned a bit further (line 106-107) in the text, carbon ions are considered high LET radiation. To make it sound less arbitrary we changed the phrasing of the sentence to a more general effect of higher LET radiation (line 128).

Line 134: The paragraphs with respect to HIF1-α expression are not clear. The only example provided is the exposure to 1GeV protons (an energy that is not clinically relevant) and which likely generates a field of additional particles (neutrons, recoil heavy ions) which will contribute to any effect. There is no indication or mechanism mentioned as to why the expression changes when protons are used. Is this because of increased complex damage which inhibits the expression? As mentioned above it is not clear what is meant by an experiment with 1GeV protons, do the protons that interact with the sample have an energy of 1GeV, is this kinetic energy, total energy? Or is the sample placed at the bragg peak where the energy is much lower and comparable to any other bragg peak.

In the referred paper it is not mentioned where in the Bragg peak profile the samples were placed. The proton energy used for the experiments is mentioned as 1 GeV, however the authors also mention that additional energies should be tested. To my knowledge, this is the only paper that investigated the expression of HIF1-α and this is thus a preliminary result. The authors checked the expression for HIF1- α as a key pro-angiogenic factor. A mechanism to why the expression is altered after photon compared to proton radiation is not mentioned, as more experiments are needed to validate the results. We have clarified this by adding the sentence at line 143-144: ‘As this is the first report about a decreased HIF-1α expression after proton irradiation, more studies are needed to be able to validate the results’ at the end of the paragraph.

Line 142: The increase in ROS of protons with respect to photons is not something that has a relationship to oxygenation but should be mentioned in the RBE section. It is likely that this is related to an increase in ionisation density, which induces more ionisations and ROS generation close to the DNA structure. Generating not only more direct damage but also a higher probability for indirect damage.

We would prefer to leave these sentences under the heading of this paragraph titled ‘Reactive Oxygen Species and Hypoxia’ (section 3) This paragraph focusses on the role of ROS and hypoxia, and the biological processes possibly influencing the RBE so we believe this is the better place to discuss this.

The RBE section (section2) is focused on the ongoing debate about the RBE of protons in comparison to photons.

Line 153: The final line does not fit here as we have not mentioned the impact the cell cycle yet. It is not clear how the previous explanation leads here. How does the ROS scavenger (also not defined ) affect the start of a cell cycle transition?

We have deleted this line (line 155-158) in this part. The paper is more thoroughly discussed in the section 4.4.

Line 159: The sentence: “ increasing LET causes more extensive and more clustered DNA damage as a result of denser ionization events”. Might be clearer as follows:

increasing LET results in denser ionisation events, which causes more extensive and more clustered DNA damage.

Thank you for the suggestion. We have changed line 164-165 according to the proposed sentence.

Line 161: This discussion might be better placed in the section on RBE or vice versa.

As mentioned before, we would like to keep the RBE section about the ongoing debate about the proton RBE in comparison to photon RBE. We think this discussion fits better in this part as here we will thoroughly discuss the biological processes that can influence the RBE.

Line 163 : How does the presence of repair enzymes make the detection of DNA damage difficult? I think I know what the authors are thinking of (i.e. the damage is repaired and therefore can’t be detected). However, the relevant damage the will lead to cell death is not repaired and should therefore still be detectable. Also one needs to be aware that the enzymatic repair process happens on a vastly different time scale compared to the molecular damage processes. But this is under the assumption that I can read minds, so the authors need to be more specific.

When staining multiple repair enzymes to check for colocalization at a site of clustered DNA damage, strong microscopes are necessary to be able to state that the proteins are indeed present at the same site of DNA damage. We have clarified this later in the paragraph (line 180-183).

Line 175: A lot of jargon starts to be used here, please define terms like histone, phosphorylation

(I know what you mean but this is an overview which should provide information to different types of specialties in research).

As the paper describes the biological unknowns of proton therapy, most of these biological processes and terminology will be familiar for the majority of the readers. But if considered suitable for the editor, we can add a separate section/footnote with a brief explanation of terms.

Line 180: Again there is the issue that the authors are not specific in what exactly is meant by proton radiation.

We have clarified what is meant with proton irradiation by adding the part ‘in the SOBP composed of 6 Bragg peaks of 100-110 MeV’ in line 195-196.

Line 190 -203: It is not clear whether γH2AX differentiates between DSB and complex cluster damage . The distinction is not clear and the terms are used in a confusing manner. Again not clear how the plateau was achieved and what the spectrum was there. One could make the statement that a the bragg peak we would get more complex damage as ionisation density would be higher.

yH2AX can not distinguish between DSBs and clustered damage. We have clarified this by adding the sentence ‘However immunofluorescent staining of γH2AX will not determine if the detected DSB is part of a clustered DNA damage site.’ (line 186-187).

Line 205: A differentiation of Double strand Breaks and complex cluster damage is made. The difference between these two is not clear.. It should be made clear how this is being determined.

In addition there is an indication that for repair purposes there is a difference between them. As indicated in the text. It is not clear how the different pathways are altered depending on the presence of clustered damage they are just mentioned.

As stated earlier in this section, clustered damage consists of multiple and different DNA lesions within a few base pairs. DSB are not clustered damage but can be one of the lesions in a clustered DNA damage site. The specific repair pathways and how these change when clustered DNA damage is present are further discussed in section 4.3. Here we first focus on the signalling of clustered DNA damage.

Line 213: Suddenly, the language in this review changes and it is only suited for a reader well versed in molecular biology. While earlier physical terms were somewhat defined, This has been thrown overboard. Many of your readers will not know what ubiquitylation is

“This ubiquitylation was catalyzed by E3 ubiquitin ligases ring finger 20/40 clustered (RNF20/40) and male-specific lethal 2 homolog (MSL2)” . This is completely unreadable, but for the initiated few.

Again we think that as the paper describes the biological unknowns of proton therapy, most of these biological processes and terminology will be familiar for the majority of the readers. However, the term ‘ubiquitylation’ can be briefly explained in a separate section/footnote if considered suitable by the editor.

Line 217: Using a neutral comet assay modified to detect and measure DSBs and oxidative clustered DNA lesions they found a linear dose response for the induction of these clustered lesions

Again not clear how the comet assay is linked to the previous part. Would one not expect an increase of lesions if the dose is increased. I fail to see the link with the previous part. Does a comet assay detect whatever you talked about. It now seems that you are able to detect DSB, as well as clustered damage and other things that was previously indicated as impossible. Further even you now say that SSB’s are repaired by PARP although we are led to believe that cell death is governed by DSB and complex damage. Again a process which is suddenly mentioned.

This comet assay can detect oxidative clustered lesions (clustered lesions including an oxidated base) but the evidence for the clustered damage here is still indirect. Clustered damage can include multiple kinds of DNA lesions, among which SSBs. PARP plays a crucial role in the repair of these SSBs. We have clarified the function of PARP in SSB repair in line 239-241.

Line 228: This part shines some light on the previous darkness. So it might be interesting to put this upfront such that you illustrate the model which is being developed and support it with data. Also this is the good place to introduce in more simple terms what the ubiquitilation process entails.

As this paragraph summarizes the findings of the studies mentioned, we prefer to leave these conclusions at the end of the paragraph.

Line 233: A good place to force a new page

Thank you for the remark. We adapted accordingly.

Line 234: This could be a more philosophical remark. While the previous part of the text has involved the generation of damage and complex processes which the DNA-molecules and cell undergo as a passive bystander . The language now changes to an active proponent where the cell becomes an active subject “it relies on signaling”? You make it seem that the cell actively decides to signal and activates the repair mechanisms. This is an antropomorphism, which unfortunately is rife in the radio-biology community. You imply that there is an intellectual process were the damage is identified and subsequently a repair pathway is chosen (on purpose) At this time there is no need for this and the remainder of this section can be well read without resorting to this.

The reviewer is referring to line 250-251: ‘When DNA damage is induced, cells rely on the DNA damage response to signal the presence of DNA damage in order to activate the correct repair mechanism.’. We rephrased this to: ‘When DNA damage is induced, DNA damage response pathways are activated in order to start the correct repair mechanism’ (line 251-252).

Line 238: This is a good place to introduce the enzymatic repair mechanisms. Specifically, those that use templates. This because as you fail to mention these templates become available when more complex damage is being introduced which could be seen as a driver of which repair mechanisms prevail. Line 244-277 then becomes more plausible.

Indeed, as we discuss in section 4.3 some research points towards a differential repair for proton induced lesions. This can be due to a competition between homologous recombination (HR) and non-homologous end-joining (NHEJ) which is mainly regulated by end-resection. NHEJ requires minimal end-resection while HR requires more end-resection to create single stranded DNA to search for its homologous template. This DNA end resection is increased clustered lesions induced by high-LET radiation. However,a definite conclusion as to whether homologous recombination plays a bigger role in the repair of proton induced lesions is not yet provided. Furthermore, as protons are not considered as very high LET, we have not further commented on this.

Line 248: Define “wild type” cells (I know what they are but others do not).

Similar to what was mentioned before, the term ‘wild type’ could be briefly defined in a separate section/footnote. But we think the majority of the readers will be familiar with the term.

Line 255: Can this be related to increased complexity? In the previous sections you have established that the difference between photons and protons is increased complexity in damage and the complexity increases downstream of a SOBP. Rather than doing that we obtain a summation of results. See also Line 277.

We have added a sentence to discuss this (line 272-274).

Line 282: A remark is provided that the results outlined above depend on several factors and that these are not disclosed adequately. For instance looking at the Carbon ion results with RBE varying from 1.07 to 2 , which leads me to presume that high energy carbon ions were used in the first. So if you propose to have controlled studies you might want to highlight what needs to be controlled and whether there could be a common language or model.

We have added some more clarification. See line 307-309.

Line 296: This sentence is difficult to read. Results are presented from the HeLa-, glioblastoma-cells and fibroblasts. The next sentence then only talks about glioblastoma-cells and fibroblasts as comparison. So either the HeLa cells were not compared or did they not show a difference. If it is the latter then the energy of the protons is important (as are the experimental circumstances) as 48MeV protons have a ionisation density comparable with photons.

The HeLa cells were not compared to photon irradiation, therefore these are not mentioned in the next sentence.

Line 304:“They found that this was the result of sustained induction of CHK2 phosphorylation and more rapid CHK1 dephosphorylation after proton radiation [57]”

None of this was previously defined and not for the uninitiated.

We have provided some more information on the results found in the referred paper for clarification (line 326-328, 330-332).

Line 313: Could you define what you mean by “expression”? Again I think I know what this but it would be great if a definition is provided. Also suddenly you speak of expression of genes after radiation, as genes of cells are altered through evolutionary processes I would not expect that they would be affected in these short terms, but rather that they are selected out? It is also the first time genes are mentioned in this paper, so a short sentence of the importance in this context would be nice.

By “expression” we mean the process of transcribing a gene into RNA (gene expression) that can be further translated to a functional protein (protein expression). When we talk about gene expression the changes in the RNA levels are considered. These can be altered after exposure to irradiation without evolutionary processes being involved.

Line 340: There is an apparent contradiction between the data presented by Miszczyk et al compared to Wang et al. This should be commented on. Different cells and different proton energies?

We have added a few sentences to comment on this contradiction in results (see line 378-381).

Line 356: Activation of pathways is now introduced. While they are well studied in radio—biology the link with differences of complexity of damage is not clear. So in this context more information on the mechanism and its relationship to damage is interesting

To our knowledge, there is limited information available on the regulation of these cell death pathways in response to clustered DNA damage. Therefore, we have not further addressed this in our manuscript. For clarification we added the sentence: ‘The cell death signalling mechanisms seem to differ between photon and proton radiation, however the relation between the signalling and the complexity of the induced DNA damage has not been clarified’ at line 393.

Line 386: Rather than hypoxia making a cell radio resistant it seems to be the other way round that oxygen makes cells more sensitive and enhances the damage.

As mentioned earlier about the role of oxygen, the absence of oxygen makes the cell more resistant to radiotherapy.

Line 394: Quite a late stage to define PARP.

We have clarified the role of PARP earlier in our manuscript (line 240-241). In this part of the paper, the function of PARP in apoptosis is mentioned.

Line 415: Usually this type of Monte Carlo is different than general purpose MC codes, This is usually considered microdosimetric monte Carlo simulations, examples are PARTRAC, MCDS, LEM IV. Also noteworthy that as the LET increases not only the ionisation density changes but also the geometry. In low LET the foci are distributed randomly through the cell nucleus, while at very large LET the follow the track line closely.

As protons are not considered as very large LET, we decided not to mention this change in foci distribution in this paper.

Reviewer 2 Report

Many congratulations to the author for this overview focusing on the biological unknowns in Proton Therapy.

I strongly believe that this extensive and careful revision of the literature will meet the interest of many clinicians, radiobiologists and researchers in this field.

I have no particular comments to this paper.

Good job!

Author Response

Thank you for taking the time to thoroughly review our work. We are pleased with your enthousiastic response.

Reviewer 3 Report

I think this paper is well-examined.

Author Response

Thank you for taking the time to thoroughly review our work and for your approval.

Round 2

Reviewer 1 Report

I found this an already much improved manuscript. I do notice some reluctance by the authors to follow some of the suggestions, which are merely given in order to increase the public and hence the citeability of the paper.

With respect to the more specialised molecular biology subjects like Ubiquitin and others, it should suffice to include a short description and a reference to more extensive works for instance:

Schwertman, P., Bekker-Jensen, S. & Mailand, N. Regulation of DNA double-strand break repair by ubiquitin and ubiquitin-like modifiers. Nat Rev Mol Cell Biol 17, 379–394 (2016). https://doi.org/10.1038/nrm.2016.58

Michael Uckelmann, Titia K. Sixma, Histone ubiquitination in the DNA damage response, DNA Repair, Volume 56, 2017, Pages 92-101, https://doi.org/10.1016/j.dnarep.2017.06.011.

In my view a short paragraph is needed explaining such concepts. As indicated in my comments to the editor. This type of review should be readable by a multi-disciplinary group, such that a more efficient experimental design can be effected in the future.

Finally, I noticed that the author order is different in the printed paper compared to the review order Verstrepen and Dok are interchanged.

Author Response

Sandra Nuyts

Full Professor Radiation Oncology

UH Leuven

3000 Leuven

Belgium

Tel:+3216347600

Fax: +3216347623

E-mail: sandra.nuyts@uzleuven.be

                                                                                                                             January 27, 2021

Dear Editor, dear Reviewer

We would like to thank reviewer 1 for his second review of our work and the constructive suggestions. We appreciate the fact that the reviewer already found the paper much improved. Reviewer 1 suggested to add additional clarification of molecular biology terms which we tried to incorporate in the manuscript. Please find below an overview of the revisions.

Sincerely,

Sandra Nuyts

Review report reviewer 1

General comments:

I found this an already much improved manuscript. I do notice some reluctance by the authors to follow some of the suggestions, which are merely given in order to increase the public and hence the citeability of the paper.

With respect to the more specialised molecular biology subjects like Ubiquitin and others, it should suffice to include a short description and a reference to more extensive works for instance:

Schwertman, P., Bekker-Jensen, S. & Mailand, N. Regulation of DNA double-strand break repair by ubiquitin and ubiquitin-like modifiers. Nat Rev Mol Cell Biol 17, 379–394 (2016). https://doi.org/10.1038/nrm.2016.58

Michael Uckelmann, Titia K. Sixma, Histone ubiquitination in the DNA damage response, DNA Repair, Volume 56, 2017, Pages 92-101, https://doi.org/10.1016/j.dnarep.2017.06.011.

In my view a short paragraph is needed explaining such concepts. As indicated in my comments to the editor. This type of review should be readable by a multi-disciplinary group, such that a more efficient experimental design can be effected in the future.

We have clarified the terms histone, phosphorylation, ubiquitylation and wild type when they were first mentioned in the text:

  • Histones: in section 4.1 Induction of DNA damage we added the sentence ‘Histones are proteins that associate with the DNA in order to pack and organize the DNA so that it fits inside the nucleus.’
  • Phosphorylation: in section 4.1 Induction of DNA damage we added: ‘… a phosphoryl group is transferred to histone H2AX in a process called phosphorylation.’
  • Ubiquitilatyion: in section 4.2 Signalling of Clustered DNA damage after Proton Irradiation we added: ‘Ubiquitylation is the post-translational process of attaching a small protein called ubiquitin to another protein. When DNA damage occurs, this process is part of the regulation of the DNA damage response, as reviewed in [82-84].’ Three references were added for further reading, including the two references suggested by the reviewer. An updated reference list is provided.
  • Wild type: in section 4.3. Differential Repair of Photon and Proton Induced Lesions we added: ‘…wild type cells (without deficiencies in HR and NHEJ)’.

Finally, I noticed that the author order is different in the printed paper compared to the review order Verstrepen and Dok are interchanged.

Thank you for the remark. We would prefer to keep the order from the manuscript: Laura Vanderwaeren, Rüveyda Dok, Kevin Verstrepen and Sandra Nuyts. This was how we intended to submit our original version and the first revision. All authors agree with this order.